# Neural Modulation for Flash Memory: An Unsupervised Learning Framework for Improved Reliability

**Jonathan Zedaka**   **Elisha Halperin**   **Evgeny Blaichman**   **Amit Berman**

Samsung Semiconductor Israel Research and Development Center

`{jonathan.z,elisha.h,evgeny.bl,amit.berman}@samsung.com`

## Abstract

Recent years have witnessed a significant increase in the storage density of NAND flash memory, making it a critical component in modern electronic devices. However, with the rise in storage capacity comes an increased likelihood of errors in data storage and retrieval. The growing number of errors poses ongoing challenges for system designers and engineers, in terms of the characterization, modeling, and optimization of NAND-based systems. We present a novel approach for modeling and preventing errors by utilizing the capabilities of generative and unsupervised machine learning methods. As part of our research, we constructed and trained a neural modulator that translates information bits into programming operations on each memory cell in NAND devices. Our modulator, tailored explicitly for flash memory channels, provides a smart writing scheme that reduces programming errors as well as compensates for data degradation over time. Specifically, the modulator is based on an auto-encoder architecture with an additional channel model embedded between the encoder and the decoder. A conditional generative adversarial network (cGAN) was used to construct the channel model. Optimized for the end-of-life work-point, the learned memory system outperforms the prior art by up to $56\%$ in raw bit error rate (RBER) and extends the lifetime of the flash memory block by up to $25\%$.

## 1   Introduction

The continued scaling of flash memory technology into smaller process nodes, combined with the increased information capacity of each flash cell (i.e, storing more bits per cell), has placed NAND flash memory at the forefront of modern storage technology. Those advances comes at the cost of increased vulnerability to various error mechanisms that can compromise data integrity and system reliability. The endurance of the flash can be measured by the number of program-erase (PE) cycles the cells can undergo, before the error-correction-code (ECC) decoder fails to reconstruct the programmed bits.

Modern flash devices suffer from multiple sources of error, such as write-induced [1], data retention [2, 3] and read-induced [4] errors. The impact of these different mechanisms vary between physical locations in the chip. In addition, the probability of errors to occur on different cells is not independent: The charge stored in a flash memory cell is affected by the state of its neighboring cells (see section 2.1.4), a phenomenon dubbed inter-cell interference (ICI).

To maintain high reliability while pushing for ever-growing storage density, manufactures devote significant resources to comprehensive characterization and optimization of flash systems [5, 6]. Over the years, the industry has developed numerous strategies for mitigating errors, including robust ECC algorithms [7], smart writing schemes [8] and adaptive read methods [9, 10]. However, all above

37th Conference on Neural Information Processing Systems (NeurIPS 2023).

techniques optimize either the data storage or retrieval operations individually, ignoring their built-in dependencies. Furthermore, they typically neglect the inter-cell correlations between the cells. To that end, we introduce the *neural-modulation* framework—an end-to-end unsupervised optimization method which is designed to mitigate the complex error patterns of flash memory devices and address the plethora of correlations between their components. By using a learning based approach, we enable (i) state-of-the-art modeling and optimization of the system; and (ii) flexibility and generalization: optimization of a new flash generation can be done in a simple and automatic procedure.

The flash memory stores information in an array of cells, where the information stored in each cell is encoded in its transistor threshold voltage $v_{th}$. Current writing schemes allocate each cell with a target $v_{th}$. The targets are optimized to result in a minimal error-rate with an emphasize on the end-of-life workpoint. So far, such prior art modulation methods base the choice of the target voltage solely on the data stored in the cell. This kind of modulation, illustrated in Fig 1, is referred to as a pulse amplitude modulation (PAM). In this work, we propose a novel approach to mitigate the ICI by optimizing the target $v_{th}$ of the memory cells with respect to the data designated to their neighbors. By employing such a scheme, we preemptively counteract the ICI.

We optimize the $v_{th}$ target modulation by performing an end-to-end training of the flash memory system (see Fig 2). Our method is based on the analogy between memory cells and communication channels—both systems receive and transmit data. Consequently, we can follow recent advances in communication research [11, 12, 13]. We use an unsupervised learning method, jointly training the transmitter and receiver of a communication system as an auto-encoder. The channel auto-encoder is trained using a reconstruction loss between the input of the encoder and the output of the decoder (see Fig. 5). In the flash memory domain, we define the transmitter of the system, which we dub the modulator, as the module translating the information bits into $v_{th}$ targets.

The joint optimization of the transmitter and receiver, using standard back-propagation, requires calculating the channel gradient with respect to the transmitted signal (see $x$ in Fig. 5). Since, however, the flash channel is non-differentiable, we cannot naively implement this technique. To overcome this issue, we train a

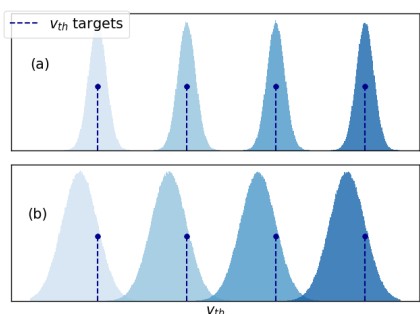

Figure 1: An illustration of a 4-PAM modulation for a Multi-Level cell NAND device (storing 2 bits per cell). Each data symbol is allocated with a *single* $v_{th}$ target that minimize the BER at the end-of life work-point. Panel (a) shows the distribution of the cells $v_{th}$ around the chosen targets immediately after the writing operation. Panel (b) illustrates the same data, at the end-of life work-point. Cells with $v_{th}$ within the overlap region between the distributions are susceptible to a read error.

conditional generative adversarial network (cGAN) based channel model as proposed in [14]. Recent work by [15] showed that GANs can be useful in modeling the $v_{th}$ distribution of the flash cells by using the data symbols. In this work, the trained channel model learns to produce samples from the $v_{th}$ distribution of a memory cell given the $v_{th}$ *targets* of the cell and its neighbors. Using the neighbors data, the channel model is able to capture the inter-cell interference statistics, allowing the modulator to counteract this phenomenon.

To reduce the additional programming latency caused by our neural modulator, we develop a novel shared pulses programming scheme. The suggested scheme, which is based on the Incremental Step Pulse Programming (ISPP) scheme [16], achieves accurate programming, without increasing the amount of programming pulses, and with only a minor increase in programming time.

When optimized for the EOL work-point, the learned memory system outperforms the prior art by up to 56% in terms of raw bit error rate (RBER), extending the tested EOL work-point of the chip by 25% in terms of PE cycles. This gain is achieved without increasing the read latency.

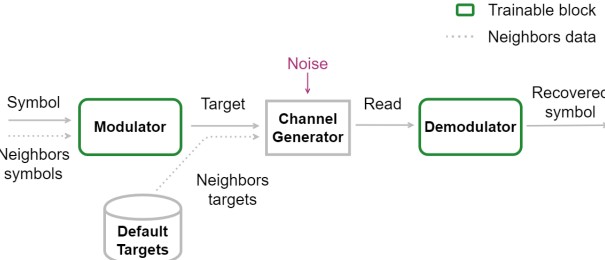

Figure 2: The neural modulation system architecture. The channel generator is a pre-trained model of the NAND device. The modulator is trained to output the optimal program $v_{th}$ target for each cell, considering its input symbol and the symbols of its neighbors. The output targets flow through the GAN based channel model that simulates the programming operation, data retention process and subsequent read operation, to output a stochastic $v_{th}$ read for the cell. From there, the $v_{th}$ read is passed to the demodulator to recover the original programmed symbol.

## 2 Preliminaries

### 2.1 Flash memory basics

#### 2.1.1 The 3D NAND flash architecture

The basic unit of a 3D-NAND Flash Memory is a **cell**—a semiconductor based charge trap transistor. The cell's information is encoded in its transistor threshold voltage $v_{th}$, which is a real value $v_{th} \in [v_{\min}, v_{\max}]$. The range $[v_{\min}, v_{\max}]$ is referred to as the **dynamic range.**

Modern flash devices typically store multiple bits per cell (**BPC**). The dynamical range is divided into $2^{BPC}$ segments, each corresponding to a different information bits dubbed **symbol**. We define a **level** as the group of cells that hold the same symbol. Flash devices that store 1/2/3/4 BPC are usually referred to as single-level cell (**SLC**)/multi-level cell (**MLC**)/triple-level cell (**TLC**)/quadruple-level cell (**QLC**), respectively.

The flash cells are organized in three dimensional arrays called **blocks** (see Fig. 3). The three dimensions of the block are the **pillar/string** axis, the **wordline** (WL) axis and the **bit-line** (BL) axis. The two dimensional cross section of the block consisting of the pillar and WL axes is called the **string select line** (SSL).

#### 2.1.2 The flash memory interface

The two most basic operations on the flash memory device are programming and reading.

1. **Programming** the block with information; and more specifically, bringing the $v_{th}$ of each flash cell to the voltage **target** assigned to the symbol that should be stored in that cell. This operation is done using the ISPP algorithm described in the supplementary material.

2. **Reading** the data stored in a WL. This operations is done by segmenting $[v_{\min}, v_{\max}]$ into $2^{BPC}$ voltage regions, corresponding to the $2^{BPC}$ data symbols.

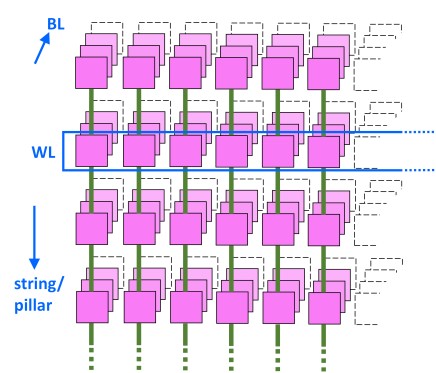

Figure 3: Top view of the 3D NAND flash block structure

### 2.1.3 Errors in NAND flash memory

Read **errors** occurs if there is a mismatch between the symbol designated for the cell and its actual threshold voltage. The fraction of bits which were incorrectly assigned by the reading algorithm out of all bits read is called the **Bit Error Rate** (**BER**).

Errors in flash memory devices can be induced by many factors [1], but for the purpose of this work we will focus only on two of them - programming variation and retention disturb.

The $v_{th}$ of a memory cell after receiving a programming pulse is a stochastic function of the initial cell $v_{th}$ and the pulse magnitude. Therefore, programming a level $\ell$ in some WL $w$ will result in some *distribution* of the $v_{th}$ close to the desired target, regardless of the writing algorithm. We define the **programming variation** as the noise injected by the programming algorithm.

The **retention disturb** [2, 3] is defined as the $v_{th}$ change during the data storage time, after the programming phase has completed. The retention disturb happens because charge leaks out of the cell over time, thus changing its $v_{th}$.

The shift in $v_{th}$ due to programming variation and retention disturb varies between WLs and SSLs. Furthermore, it is also a function of the number of PE cycles experienced by the block. Each PE cycle wears down the block cells, making them more vulnerable for disturbs, and therefore increases the BER of the block.

### 2.1.4 Inter-cell interference in the 3D-NAND block

The most dominant coupling between cells in a 3D-NAND, and hence Inter-Cell Interference (ICI), occurs along the pillar axis, and it includes phenomena such as programming interference and lateral migration. The programming interference is the phenomenon at which programming of a flash cell increases the $v_{th}$ of adjacent neighbor cells [1]. The programming interference effect on a flash cell is a stochastic function of the amount and magnitude of the pulses its neighbors experience. Therefore, it is also a function of the $v_{th}$ *targets* of its neighbors.

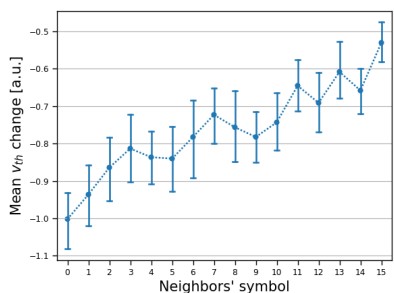

The lateral migration [1, 17], which takes place during data retention period, describes charge migration between cells with different $v_{th}$. Fig 4 illustrates the lateral migration phenomenon measured on a 3D-NAND chip programmed with QLC data. We measure the $v_{th}$ of each cell immediately after programming the block, and again after an effective retention period of one month at room temperature.[1] The figure

Figure 4: Lateral migration out of a 3D-NAND memory cell as a function of its neighbor symbol. The change of $v_{th}$ of level 15, shown in the figure, was measured on a single WL of a QLC flash memory device. The charge loss grows as the neighbors' symbol is lower. The figure is presented with 95% confidence interval.

shows the mean $v_{th}$ change of the cells programmed to data symbol 15 as a function of the symbol of its two adjacent cells. We obtained that low-symbol-neighbors cause higher voltage loss over time.

## 2.2 Generative adversarial networks

Generative adversarial networks (**GAN**s) [19] are a class of generative models consisting of two separate neural networks (NNs), a generator $G_{\Theta}$ and discriminator $D_{\Psi}$. The basic setup of GANs consists of a set of samples $\{x_i\}_{i=1}^m \subset \mathcal{X}$ drawn from some distribution of interest $p$, which we wish to produce new samples from. GANs learn the distribution $p$ by fixing a rather simple distribution $p_{\mathcal{Z}}$ (Gaussian for example) over a latent space $\mathcal{Z}$ and then optimizing the generator function $G_{\Theta} : \mathcal{Z} \to \mathcal{X}$ such that the push-forward measure $p_G := G_{\star}(p_{\mathcal{Z}}) = p_{\mathcal{Z}} \circ G^{-1}$ will be as close as possible to $p$.

---

[1]The effective data retention time is simulated at $100°C$ according to Arrhenius' Law as described in [18]

In this work we apply a conditional version of the Wasserstein GAN (WGAN) proposed in [20, 21], which uses the Wasserstein distance as the training loss of the generator. WGANs were proven to enable more stable training and convergence[20].

### 2.3 End-to-end training of autoencoder-based communication systems

End-to-end training of autoencoder-based communication systems is a relatively new approach [13] for the joint optimization of both the transmitter and the receiver of the system. With this approach, an autoencoder-like architecture is used, where the transmitter acts as the encoder and the receiver as the decoder. Both are represented as NNs with parameters $\Theta_T$ and $\Theta_R$, respectively, which we train *jointly* using back-propagation. The training uses a reconstruction loss between the symbol $s$ at the input of the transmitter, and the reconstructed symbol $\hat{s}$ at the output of the receiver (see Fig 5), where

$$\hat{s} = f_{\Theta_R}\left(C\left(f_{\Theta_T}\left(s\right)\right)\right)$$

This reconstruction loss is commonly defined as the categorical Cross-Entropy (CE) between $s$ and $\hat{s}$, where $C$ represents the system channel.

## 3 End-to-end learning based modulation for the flash memory

The basic program-read flow of the 3D-NAND flash memory suffers from multiple sources of error, degrading its endurance both in terms of PE cycles, as well as data retention time. To mitigate the effects of those errors we introduce a *neural-modulation* system, which is inspired by the end-to-end training approach described in Section 2.3. Given some programming scheme, the neural-Modulation system optimizes the $v_{th}$ targets of the 3D-NAND block. On top of optimizing the $v_{th}$ targets, our system performs *pre-distortion* of the target of each cell, to counter the ICI phenomenon (see section 2.1.4). For this purpose, the $v_{th}$ target is optimized *conditioned on the symbols of its pillar neighbors*. This novel approach gives rise to a significant BER reduction, and by that, extend the EOL work-point of the NAND flash block.

### 3.1 Flash memory as classical communication system

In order to fully describe the end-to-end training approach outlined in the previews section, we first turn to formally define the NAND flash memory as a classical communication system. The three components of a classical communication system are the transmitter, channel and receiver. We adjust their definition to memory devices in the following way:

1. The transmitter $f_{\Theta_M}$, which we dub the **modulator**, is the component responsible for translating the input symbols $s$ into $v_{th}$ targets for the programming algorithm.

2. The channel $C$ is the composition of the programming scheme, the data retention process, and the $v_{th}$ read operation. It simulates the disturbs and errors described in Section 2.1.3.

3. The system receiver $f_{\Theta_D}$, which we dub the **demodulator**, is the component which reconstructs the data symbols $\hat{s}$ from the $v_{th}$ readings.

Using this formulation we can adopt the end-to-end learning based approach described in Section 2.3 to learn an optimal modulation for our channel. Bellow we elaborate on the three main components of our system.

### 3.2 cWGAN as NAND channel model

We model our NAND channel as a cWGAN [21]. Modeling the NAND as a cGAN is essential for training the auto-encoder because it allows differentiating the channel [14]. The generator is trained

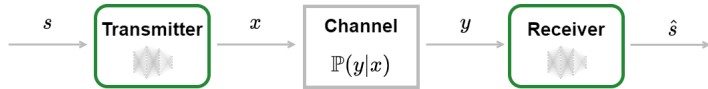

Figure 5: The structure of an end-to-end learning based communication system.

to learn the cell's threshold voltage $v_{th}^{cell}$ distribution given its "context" - the cWGAN condition - which includes the following:

1. The program target of the cell $t^{cell}$.
2. The target $v_{th}$ of the two nearest pillar neighbors of the cell $T^{neighbors} \in \mathbb{R}^2$.
3. The SSL index, $ssl \in [N_{ssls}]$ and WL index $wl \in [N_{wls}]$ of the programmed WL.

Including $T^{neighbors}$ in the cell context is crucial for modeling the ICI effects. Consequently, it allows the neural-modulator to adjust the target of each cell according to $v_{th}$ of its neighbors and counteract the ICI. Since different WLs in the block exhibit different reactions both to programming pulses as well as the ICI noise, the SSL and WL indices of the programmed WL are also used to condition our cWGAN. We can therefore formulate the cWGAN objective as learning to produce a sample of $v_{th}^{cell}$ ($v_{th}$ reads from the NAND) given the conditional density

$$\mathbb{P}\left(v_{th}^{cell}|t^{cell}, T^{neighbors}, ssl, wl\right)$$

For full model specification see the supplementary material.

### 3.3 The neural modulator and demodulator

The next two components of our system are the modulator and the demodulator. The modulator is a function

$$f_{\Theta_M} : \left[2^{BPC}\right] \times C \longrightarrow [v_{min}, v_{max}],$$

mapping *information symbols* $s \in \left[2^{BPC}\right]$ to $v_{th}$ targets to be programmed to the NAND, conditioned on the "context" of the target cell $c \in C$. The context $c$ comprises of the symbols designated to the two neighboring cells along the SSL (one on each side of our given cell), the SSL index and WL index of the target cell $C = \left[2^{BPC}\right] \times \left[2^{BPC}\right] \times [N_{ssls}] \times [N_{wls}]$.

The demodulator,

$$f_{\Theta_D} : [v_{min}, v_{max}] \times [N_{ssls}] \times [N_{wls}] \longrightarrow \mathbb{R}^{2^{BPC}},$$

receives a $v_{th}$ read from the channel. It then reconstructs the programmed symbol by learning the log likelihood ratio (LLR) of each symbol $s \in \left[2^{BPC}\right]$ given the $v_{th}$ read. Note that the demodulator does **not** receive any information about the cell's neighbors. This forces our system to exploit the information about the neighbors only at the modulation stage, and thus perform pre-distortion to offset the ICI. In other words, all the information about the neighbors is already encoded by the modulator into its output. This independence of the demodulator from the cell's neighbors also guarantees that our system will not require any changes to be made to the current reading algorithm. As there are physical differences between different sections of the block, we use the SSL and WL indices as input to both the modulator and the demodulator.

The number of layers and other NN parameters of both modulator and demodulator are given in the supplementary material. The output of the modulator is passed through a $\tanh$ function, to enforce it to be in the range of $(-1, 1)$. This output is then scaled to the dynamic range of the chip $(v_{min}, v_{max})$, forcing the modulator to choose only targets that are feasible for programming.

### 3.4 The training procedure

The Neural-modulation training procedure is divided into two steps. In the first step we train the cWGAN based channel model. The data we use as a train set is a set of $v_{th}$ reads of a single block programmed with given targets $T$.[2] The critic is utilized solely to train the generator, and is afterwards discarded. The second step includes the joint training of the modulator and demodulator. As a prerequisite, we assume to have the pretrained, frozen channel model (i.e. the generator from the previous step). At each iteration a batch $B$ of samples $b \in \left[2^{BPC}\right] \times C$ flows through the modulator, generating a program target for each sample. At this point each cell has a corresponding target, but we still require the targets of its neighbor cells as well, to be used as an input for the channel model. We choose the neighbor cells targets according to some constant set of $2^{BPC} - 1$ targets that we spread uniformly over the dynamic range $[v_{min}, v_{max}]$ (we refer to those targets as the "default

---

[2]We use a set of $2^{BPC} - 1$ targets that we spread uniformly over the dynamic range $[v_{min}, v_{max}]$

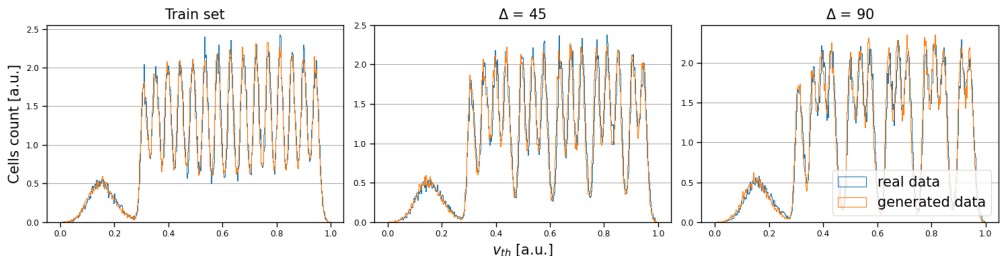

Figure 7: Comparison of the real and generated $v_{th}$ histograms of a single QLC WL.

targets"). The targets and neighbor targets then pass through the channel model to sample from the $v_{th}$ distribution of the cells when programming them to those targets. Finally, the demodulator is used to predict the original symbols given the simulated $v_{th}$ and the SSL and WL indices of the cells. As a training loss we use the CE loss between the recovered symbols and the input symbols as described in Section 3. The supplementary material provides the full details of all training parameters.

### 3.5 Programming the learned targets

The modulator, which translate a block of information symbols into $v_{th}$ targets, is trained upon completion of the second stage. However, the number of target values suggested by the system is significantly larger than the number of data symbols, as it is equal to the number of possible values of the input. As the number of unique targets increases, programming operation latency increases, making it impractical with the conventional programming scheme. Therefore, we propose performing the following two steps:

1. Quantization of the modulator output, by grouping together close targets to a single target.

2. Expanding the ISPP program scheme, to enable it to program simultaneouslymultiple, relatively close, targets, using a shared pulses mechanism. The proposed scheme preserves the current number of programming pulses and introduces solely read operations. As per [22], the effect of these read operations on the programming latency is determined to be negligible.

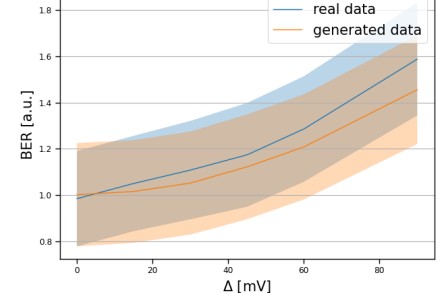

Figure 6: The real vs generated BER as function of the targets shift. The shaded area marks the standard deviation of the tested WLs.

Using these steps, we are able to program learned targets, while still maintaining a state-of-the-art programming time. A detailed description of both steps is given in the supplementary material.

## 4 Experiments

One of the main goal of the neural-modulation system is to extend the end-of-life work-point of the chip in terms of PE cycles and data retention time. Therefore, we conduct our experiments at a work-point with high amount of errors, to simulate an EOL scenario.

### 4.1 GAN channel model experiments

To evaluate the channel model, we first train our cWGAN on a $v_{th}$ read of a single block programmed with 15 linearly spread targets (for QLC modulation) over the dynamic range. We then evaluate the

trained model over a test set which include six blocks. In each test block, we perturb the linearly spread targets $\{T_i^{train}\}_{i=1}^{15}$ by setting $T_i = T_i^{train} + \lambda_i \Delta$, where

$$\lambda_i = \begin{cases} 0 & i \in \{1,3,6,9,12,15\} \\ 1 & i \in \{2,5,8,11,14\} \\ -1 & i \in \{4,7,10,13\} \end{cases}$$

for $\Delta$ in $\{15,30,45,60,75,90\}$. The evaluation itself uses two criteria:

1. BER: We compare the real and generated BER of each WL in the block.

2. Histograms: We compare between the histograms generated by our model and those measured on a real chip (i.e. when programming an actual batch of cells to the same target).

The quality of both train and perturbed predicted $v_{th}$ histograms can be seen in Fig. 7. Specifically, we measure a Wasserstein distance of $6 \pm 2$ mV between the real and generated histograms. For comparison, the average Wasserstein distance between two real blocks in the same chip is 6 mV as well. A detailed description of the distance calculation is given in the supplementary materials. In Fig. 6 we compare the real and predicted BER of a QLC block for different target voltages, i.e., values of $\Delta$. Again, we see that the BER prediction is very good and has an error margin of up to 5%. This reassures us we can indeed use the trained model to generate targets from the entire $v_{th}$ range (including targets it did not train over).

## 4.2 Neural-modulation experiments

We train the neural-modulation system as described in Section 3.4. Table 1 summarizes the performance of the model in terms of average block BER at the EOL work-point. We note that not only did the model significantly improve the BER compared to the baseline targets, but it also accurately predicted the BER results via the channel model GAN (see the 'GAN prediction' column of Table 1). Specifically, we pass the modulator targets through the GAN, and use the generated $v_{th}$ distribution to calculate the predicted BER.

Fig. 8 illustrates the pulse amplitude modulation (PAM) constellation the modulator learned for a single QLC WL, compared to the original 16 PAM constellation. The modulator found lower targets for cells with higher symbol neighbors. Such scheme is consistent with the observation discussed in Section 2.1.4, that the charge lose depends on symbols of the cell and its neighbors.

Next, we analyze the model effect as function of the data retention time. We measure the average BER as a result of programming with the default and modulated targets, at different time points during the data retention process. Fig. 9 (a) presents the BER reduction of the modulated targets compared to the default targets. We found that the pre-distortion of the targets helps keeping the BER low over

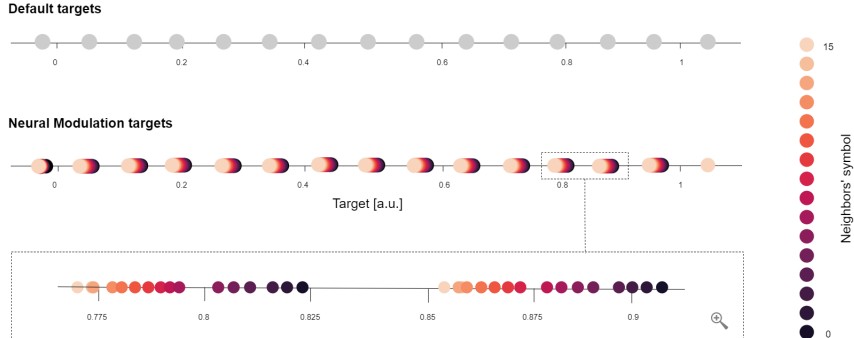

Figure 8: The original and learned PAM constellations. The first row presents the original targets of the WL which we used for the channel model training. The second row shows the learned by the neural modulator; a zoom in of symbols 12 and 13 is displayed in the bottom row. In agreement with the observation from Section 2.1.4, the modulator found lower targets for cells with higher symbol neighbors, as we saw that such cells tend to lose *less* charge over time.

Table 1: The BER reduction achieved by the neural modulator. We extract the redaction by calculating the BER after programming the cells once with the targets found by the neural modulator and once with the default targets.

| BPC | Average | | Worst case WL | |
|---|---|---|---|---|
| | GAN prediction | Empirical result | GAN prediction | Empirical result |
| QLC | 16% | **17%** | 8% | **9%** |
| TLC | 54% | **56%** | 35% | **36%** |

longer retention periods. Interestingly, the pre-distortion slightly deteriorate the BER at short times. Since the BER at these time is overall small, there is no real price for using our modulator.

The ultimate purpose of our modulator is to extend the EOL work-point of the NAND block. We would, therefore, like to estimate the change in the number of PE cycles before the the memory device reaches its EOL. The EOL point is determined by a threshold BER value above which the data cannot be reconstructed. This information can be extracted from Fig. 9 (b), where we show the BER after a constant retention time as a function of PE cycles. We found that the modulator extends the number of PE cycles by 25% (notice that the BER is normalized to its EOL value)

## 5   Conclusions and limitations

In this work, we demonstrated the potential of pre-distorted programming of the flash memory device to mitigate errors caused by ICI. We applied state-of-the-art optimization methods for the programming $v_{th}$ targets by performing end-to-end training of the channel's modulator and demodulator through a pre-trained channel model. To the best of our knowledge, our work is the first to introduce an unsupervised learning based modulation for the flash memory. The importance of our modulator is in reducing the BER for the stored information by 56% compared to conventional writing schemes, and extending the NAND EOL work point by 25% in terms of PE cycles. Currently, our modulator was optimized for a single work-point with a constant read disturb and PE cycles. Future work may generalize the channel model and modulator to overcome this limitation using a dedicated input features. Another drawback is that implementing the modulator in a flash controller requires allocating extra write buffers for the neighbors data.[3]

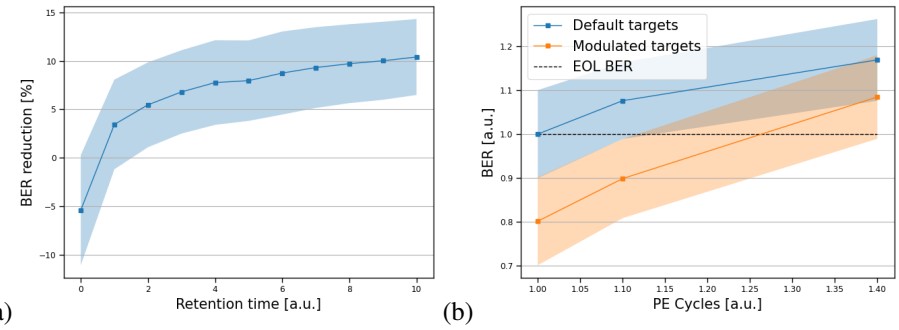

(a)                                                                                              (b)

Figure 9: Comparison between the default and modulated targets BERs. Panel (a) shows the model's BER reduction at different time points during the data retention process, measured on QLC flash. The modulated targets BER is higher right after programming the block, and lower along the rest of the retention process due to the effect of the ICI on the programmed cells. Panel (b) presents the block's BER at different PE cycle workpoints and effective data retention time of one month at room temperature. Programming the block with the modulator generated targets extends the EOL point of the block by 25% in terms of PE cycles, while maintaining the same BER. The shaded area marks the standard deviation of the tested WLs.

---

[3]Modern programming schemes already compromise about the buffer size for increased programming accuracy [22].

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
