# A    Incremental step pulse programming

The Incremental Step Pulse Programming (ISPP) scheme [1],[2, pp. 72-74] described in Alg. 1 is an iterative scheme for programming a group of cells from a single WL to a given $v_{th}$ target. The algorithm is composed of two basic operations:

1. The **programming pulse** is an operation that increases the $v_{th}$ value stored in flash cell**s**. It can be applied to a sub-set of cells $I$ from a single WL $w$.

2. **Verify** is an operation that checks whether the threshold voltages of cells from a given WL are above a certain reference voltage $t$, or below it. Formally, if $v_{th}$ represents the vector of threshold voltages stored in the cells of WL $w$, the output of the verify function $f_{verify}$ for the $i_{th}$ cell in the WL is

$$[f_{verify}(v_{th}, t)]_i = \begin{cases} 1 & v_{th}^i < t \\ 0 & v_{th}^i \geq t \end{cases}$$

The ISPP algorithm receives an inhibit vector $I$ where $I_i = 0$ if we want to program the $i_{th}$ cell of the WL and $1$ otherwise. Considering the inhibit vector $I$, the algorithm performs a series of at most $M$ pulses starting from a pulse with magnitude $v_{pgm\_start}$, where in each iteration it increases the pulse magnitude by $\Delta_{pgm}$. After each pulse a verify operations at $v_{verify}$ is applied, and all the cells above $v_{verify}$ are inhibited, i.e., they will not be affected by the pulse at the next step. We note that the ISPP parameters $v_{pgm\_start}$, $\Delta_{pgm}$ and $v_{\text{verify}}$ are a function of the $v_{th}$ target.

---

**Algorithm 1** ISPP
---
**Input** inhibit vector $I^0$ and ISPP parameters $v_{pgm\_start}$, $\Delta_{pgm}$ and $v_{\text{verify}}$.

1. **for** $j$ in $[M]$ **do**

   (a) $v_{\text{pgm}}^j = v_{\text{pgm\_start}} + j * \Delta v_{\text{pgm}}$

   (b) apply a prog. pulse with parameters $v_{\text{pgm}}^j$, $I^j$

   (c) apply verify operation at voltage $v_{\text{verify}}$, and denote result as $f_{verify}^j(v_{verify})$

   (d) create an updated inhibit vector $I^{j+1}$ where $I_i^{j+1} = \neg f_{verify}^j(v_{verify})_i | I_i^0$, meaning that cell $i$ was *not* inhibited in $I^0$ and also *below* $v_{verify}$.

2. **end for**

---

# B    Programming the learned targets

The following sub-sections describe the two steps we perform in order to reduce the additional programming latency caused by our neural modulator.

## B.1    Targets quantization

The quantization step is done by clustering the set of targets the modulator produces for each WL into $Q := 4 * (N_{Levels} - 1)$ groups, where $N_{Levels}$ is 16 for QLC flash and 8 for TLC flash. We cluster the targets using a one dimensional K-means model with $Q$ clusters. Let $T_{wl}^q$ be the set of cluster means. Once all the targets are clustered, each target $t_{cell}$ the modulator outputs is replaced by the mean of the cluster it belongs to, becoming the new cell target $t_{cell}^q \in T_{wl}^q$. We note that the one dimensional K-means problem can be solved using an exact polynomial time dynamic programming algorithms [3, 4].

The number of quantized targets was a compromise between two important considerations - the WL programming time and the quantization loss. Fig. 1 shows the BER achieved by the quantized QLC neural modulator as a function of the number of quantization levels. We see that programming 60 targets, which is equivalent to ~4 targets per symbol (for QLC modulation), preserves most of the gain of the model, while avoiding most of the programming overhead of the full (non-quantized) target range.

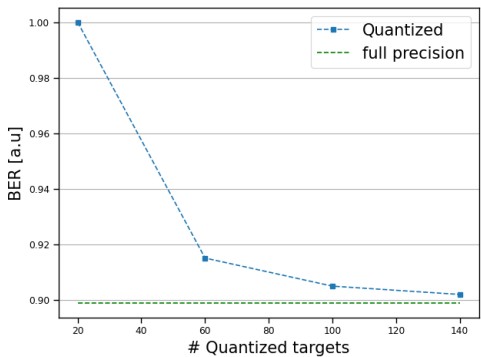

Figure 1: The BER achieved by the quantized QLC modulator, as a function of the number of quantization levels. We see that quantizing the modulator output to 60 targets preserves most of the gain of the model.

## B.2 Shared Pulses Incremental Pulse Programming

After the quantization step, we are left with a set of $Q$ program targets for each WL. Since the $v_{th}$ difference between consecutive targets is relatively small, we can use the same programming pulses to simultaneously program several targets at once. Similar to the ISPP algorithm (See appendix A), our programming scheme, which is fully described in Alg 2, uses incremental step pulse programming to program each symbol. However, since we now program cells to several different targets at once, we need to distinguish between cells with the same symbol but different targets. To that end, our algorithm performs extra verify operations after each pulse - one for each target we program together. This means that if the quantizer chose to allocate four targets for a certain symbol we will perform four verify operations after each pulse (instead of the single verify of the ISPP). As the proposed scheme *shares* all pulses between targets of cells from the same symbol, it does not increase the number of pulses required to program each WL. Note that the latency overhead of performing successive verify operations with close $v_{\text{verify}}$ values, is small compared to a single verify operation [5], which ensures our scheme preserves the original programming latency.

---

**Algorithm 2** ISPP with shared pulses

---

**Input** A list of $v_{th}$ targets $T$ that share pulses between them, a list of inhibit vectors $I^0$ , where each $I_t^0 \in \{0, 1\}^{size(wl)}$ is the inhibit vector for target $t \in T$. The ISPP parameters $v_{pgm\_start}$, $\Delta_{pgm}$ and $v_{\text{verify}}$.

    1. **for** $j$ in $[M]$ **do**

        (a) $v_{\text{pgm}}^j = v_{\text{pgm\_start}} + j * \Delta v_{\text{pgm}}$

        (b) Create a unified inhibit vector $I = I_1^j \& \ldots \& I_t^j$, i.e. all the cells which are not inhibited by one of the inhibit vectors.

        (c) Apply a prog. pulse with parameters $v_{\text{pgm}}^j$, $I$

        (d) **for** $t$ in $T$ **do**

            i. apply verify operation at voltage $t$, and denote result as $f_{verify}(t)$

            ii. set $I_t^{j+1} = I_t^j | \neg f_{verify}(t)$, i.e. add cells that reached the target to their inhibit vector.

        (e) **end for**

    2. **end for**

---

## C   Wasserstein distance

### C.1   Definition

The Wasserstein distance, is a distance function defined between two probability measures over a given metric space. Formally, given a metric space $(M, d)$ and two probability measures $\mu$ and $\nu$ over $M$, the $p_{th}$ Wasserstein distance between $\mu$ and $\nu$ is defined as

$$
\begin{aligned}
W_p\left(\mu, \nu\right) & := \\
& = \left(\inf_{\gamma \in \Gamma(\mu, \nu)} \int_{M \times M} d\left(x, y\right)^p d\gamma\left(x, y\right)\right)^{1/p} \\
& = \left(\inf_{\gamma \in \Gamma(\mu, \nu)} \mathbb{E}_{(x,y) \sim \gamma}\left[d(x, y)^p\right]\right)^{1/p}
\end{aligned}
\tag{1}
$$

where $\Gamma\left(\mu, \nu\right)$ denotes the set of probability measures over $M \times M$ with marginals $\mu$ and $\nu$. The intuition to the above definition comes from the optimal transport problem - $\gamma\left(x, y\right)$ indicates how much "mass" must be transported from x to y in order to transform the distributions of $\mu$ into the distribution of $\nu$.

In this paper we measure the 1-Wasserstein distance which is also known as the Earth-Movers distance. Therefore, from now we will use $W\left(\mu, \nu\right)$ instead of $W_1\left(\mu, \nu\right)$

### C.2   Empirical estimation of the Wasserstein distance in the $1$D case

In the special case where $\mu$ and $\nu$ are probability measures over $\mathbb{R}$, the Wasserstein distance has an analytic solution

$$
W\left(\mu, \nu\right) = \int_0^1 \left(\left|F_\mu^{-1}(z) - F_\nu^{-1}(z)\right| dz\right)
$$

where $F_\mu\left(z\right)$ and $F_\nu\left(z\right)$ are the cumulative distribution functions of $\mu$ and $\nu$. This closed form allow us to estimate the Wasserstein distance between $\mu$ and $\nu$ given $n$ drawn samples from each distribution. The $m_{th}$ element in a *sorted* sample from $\mu$ approaches $F_\mu^{-1}\left(m/n\right)$ when $n \to \infty$. With that we can easily use numerical integration to evaluate the above integral.

## D   Model parameters

### D.1   cWGAN Channel model

The model and training configuration of the cWGAN NAND channel model are presented in Table 1. Both generator and critic are multi-layer perceptron (MLP) NNs. Each trainning step is composed of five critic updated followed by a single update of the generator.

### D.2   Neural modulator and demodulator

The neural modulator and demodulator parameters are presented in Table 2. Both the modulator and the critic are MLP NNs with the same hidden layers and activations.

Table 1: The exact parameters of the cWGAN channel model

| parameter | value |
|---|---|
| learning rate | $5 \cdot 10^{-5}$ |
| batch size | 20000 |
| generator hidden layer sizes | $[128, 64, 64, 16]$ |
| generator condition input size | 5 |
| generator nz (number of noise inputs) | 1 |
| noise input (z) | $\mathcal{N}(0, 1)$ |
| generator out layer size | 1 |
| generator activation | $\tanh$ |
| critic hidden layer sizes | $[128, 128, 64, 64, 16]$ |
| critic input layer size | 6 |
| critic out layer size | 1 |
| critic activation | $ReLU$ |
| gradient penalty coefficient | 0.1 |
| optimizer | Adam |

Table 2: The exact parameters of the neural-modulation system

| parameter | value |
|---|---|
| learning rate | $5 \cdot 10^{-5}$ |
| batch size | 5000 |
| modulator hidden layer sizes | $[128, 128, 64]$ |
| modulator input layer sizes | 5 |
| modulator out layer size | 1 |
| modulator activation | $ReLU$ |
| demodulator hidden layer sizes | $[128, 128, 64]$ |
| demodulator input layer size | 3 |
| demodulator out layer size | 1 |
| demodulator activation | $ReLU$ |
| optimizer | Adam |