# OpenReview forum: "Neural Modulation for Flash Memory: An Unsupervised Learning Framework for Improved Reliability"
_NeurIPS.cc/2023/Conference — NeurIPS 2023 poster_

### Official Review · Reviewer_zVdi · 2023-06-17

**Soundness:** 2 fair
**Presentation:** 3 good
**Contribution:** 3 good
**Rating:** 7
**Confidence:** 5

**Summary:**

A neural modulator is proposed and sufficient results are presented.

**Strengths:**

Sufficient results are presented.

**Weaknesses:**

No

**Questions:**

1. What kind of noise added in the figure 1.
2. Elaborate briefly about the neural modulation comparison with conventional modulation.
3. Explain the experimental setup.

---

> ### Author Rebuttal · Authors · 2023-08-09
>
> Dear reviewer,
>
> Thank you for the time and effort you have dedicated to reviewing our manuscript.
> Below are detailed answers to your questions:
>
> Q: What kind of noise added in the figure 1.
>
> A: Figure 1 is only an illustration of a real Vth distribution. We used a gaussian noise to generate the levels.
>
>
> Q: Elaborate briefly about the neural modulation comparison with conventional modulation.
>
> A: Conventional modulation schemes assign each memory cell with a Vth target based only on its designated data symbol. Our neural modulation, on the other hand, assigns each cell a Vth target, based both on its data symbol, the WL index, and the data symbols designated to its pillar neighbors. By taking into account the data of the neighbor cells, our proposed modulation preemptively addresses the inter-cell interference (ICI) between adjacent cells in the block. The ICI is one of the main sources of error in flash memory devices, and, therefore, our modulator allows for a significantly lower error-rate at the end-of-life work point.
>
>
> Q: Explain the experimental setup.
>
> A:  The data collected from a real, currently in mass production, VNAND flash memory devices chips (TLC as well as QLC) is used both for training and evaluating the model. Specifically, the experimental data is used for calculating the loss. Due to the nature of our model, the data has to be measured at the end-of-life work-point and after long retention time. For this purpose, we perform a series of programming and erase cycles, to wear the block and get to an end-of-life work-point, before starting the data retention period. The long retention period is simulated by heating the device to 100 deg C. The threshold voltage is read with resolution of 20 mV at room temperature.

---

### Official Review · Reviewer_8xzT · 2023-07-04

**Soundness:** 3 good
**Presentation:** 3 good
**Contribution:** 2 fair
**Rating:** 7
**Confidence:** 4

**Summary:**

The paper introduces a programming method for 3D NAND flash memory. In 3D NAND, data is stored in the voltage level of cells. A cell can typically store a few bits of information (2,3,4 etc). As device density increases, the data is prone to multiple types of errors. The paper is targets two types of errors, those caused by inter-cell interference, and those caused by charge loss in time (retention error). The authors propose a machine learning scheme, based on an auto-encoder architecture with an additional cGAN (conditional generative adversarial network) between the encoder and the decoder. The auto-encoder model has been proposed before for communication systems. The addition of the cGAN is novel, and it is used to model the channel noise. The cGAN is used in order to overcome the fact that the flash channel is non-differentiable. The goal of the scheme is to predict better voltage thresholds for programming (data writing), by taking into account a more informative context for each cell (the value of its neighbors along the vertical channel dimension, and the position in the block). The new model is trained on data coming from on block of NAND and tested on data from 6 other blocks. Experimental evaluation shows a reduction of bit error rate by 56% and an extension of the program-erase cycles

**Strengths:**

The topic is good application of modern machine learning tools the a specific problem of data storage in 3D NAND flash memory. The presentation is very clear, and the paper offers a good amount of details and background about all the components of the scheme.

**Weaknesses:**

The paper has a very specialized focus, and the techniques that are presented may have limited impact. The experimental assessment is relatively small.

**Questions:**

1. It seems not clear whether any part of the experimental evaluation was performed on a physical device, or it was simulated. Could you please give some details?

2. It seems that when you move to a new block, the SSL would be a new physical location. I'm wondering if it is useful to include this value in the context. In a similar vein, the ICI in 3D NAND is characterized in various sources, for example the one cited below. As we see there, of the two immediate neighbors in the pillar, the one that was programmed before has small ICI, while the one that will be programmed after has high ICI. Similarly, the neighbors along the logical wordline have significant ICI. Would it help to include this information, perhaps in a weighted fashion in your context?

S. K. Park and J. Moon, "Characterization of Inter-Cell Interference in 3D NAND Flash Memory," in IEEE Transactions on Circuits and Systems I: Regular Papers, vol. 68, no. 3, pp. 1183-1192, March 2021, doi: 10.1109/TCSI.2020.3047484.

3. In a 3D NAND there are potentially thousands of blocks. Do you have a proposal for scaling your scheme, how should the training be performed?

---

> ### Author Rebuttal · Authors · 2023-08-09
>
> Dear reviewer,
>
> We appreciate your thoughtful feedback on our work. Below are detailed answers to your questions:
>
> Q: It seems not clear whether any part of the experimental evaluation was performed on a physical device, or it was simulated. Could you please give some details?
>
> A: We used data collected from real, currently in mass production, VNAND flash memory chips (TLC as well as QLC) for training and evaluating the model. Thus, while the channel model is generated by the GAN, the calculation of the loss is performed on actual experimental data.
>
> Due to the nature of our model, the data has to be measured at the end-of-life work-point and after long retention time. For this purpose, we perform a series of programming and erase cycles, to wear the block and get to an end-of-life work-point, before starting the data retention period. The long retention period is simulated by heating the device to 100 deg C. The threshold voltage is read with resolution of 20 mV at room temperature.
>
> As part of the channel model evaluation experiment, we trained our model over data from a single block, and tested our model over six, unseen blocks. Overall, the training set includes several hundred WLs and the testing set several thousand WLs.
>
> Q: It seems that when you move to a new block, the SSL would be a new physical location. I'm wondering if it is useful to include this value in the context. In a similar vein, the ICI in 3D NAND is characterized in various sources, for example the one cited below. As we see there, of the two immediate neighbors in the pillar, the one that was programmed before has small ICI, while the one that will be programmed after has high ICI. Similarly, the neighbors along the logical wordline have significant ICI. Would it help to include this information, perhaps in a weighted fashion in your context?
>
> S. K. Park and J. Moon, "Characterization of Inter-Cell Interference in 3D NAND Flash Memory," in IEEE Transactions on Circuits and Systems I: Regular Papers, vol. 68, no. 3, pp. 1183-1192, March 2021, doi: 10.1109/TCSI.2020.3047484.
>
> A: In the specific NAND devices we work with, the effect of the neighbors along the logical WL on the target cell is negligible compared to the two neighbors along the pillar. Therefore, in our case, the gain of using those neighbors as part of the context vector is very small, while the implementation overhead (in real NAND device) is very high. Generally, adding the WL neighbors to the context of the model is an easy and trivial extension of our work, and can be easily implemented in flash devices in which the inner WL correlations are significant.
>
> It is important to note that the variation between blocks in the same NAND device are small in our system. In particular, the similarity between the SSLs in different blocks is prominent. The physical location of the block may have a more significant effect on physical properties in other NAND devices. For such system we will have to introduce a new feature which account for the physical location and/or the mapping between the SSL on different blocks. This is possible as long as there is a continuous dependence on location, or, alternatively, as long as blocks can be characterized into a few classes.
>
> Q: In a 3D NAND there are potentially thousands of blocks. Do you have a proposal for scaling your scheme, how should the training be performed?
>
> A: As part of the channel model evaluation experiment, we trained our model over data from a single block, and tested our model over different six blocks. We found that the model predicts the Vth histograms very well both in terms of the BER, and histogram similarity (see lines 285-300). In general, we consistently find that the variation between blocks with similar conditions (such as PE cycles) are relatively small. To scale our scheme for different kind of blocks (such as blocks with High/Low PE cycles) we can extend both the modulator and channel model’s context, to include the PE cycle of the block as an input parameter. Using this extra feature will allow the system to optimize the programming operation for the specific block state.

---

> > ### Comment · Reviewer_8xzT · 2023-08-12
> >
> > Thank you for the detailed answers. I will increase my rating from 6 to 7.

---

### Official Review · Reviewer_EdAW · 2023-07-06

**Soundness:** 3 good
**Presentation:** 3 good
**Contribution:** 3 good
**Rating:** 6
**Confidence:** 4

**Summary:**

This paper presents a novel approach for improving the reliability of NAND flash memory by utilizing unsupervised learning and neural modulation techniques. The authors constructed and trained a neural modulator that reduces programming errors and compensates for data degradation over time. The proposed framework outperforms prior art by up to 56% in raw bit error rate (RBER) and extends the lifetime of the flash memory block by up to 25%.

**Strengths:**

a. Overview:
The study proposes a novel approach for modeling and preventing errors in NAND flash memory using generative and unsupervised machine learning methods. The authors utilize a neural modulator that translates information bits into programming operations on each memory cell, reducing programming errors and compensating for data degradation over time. The modulator is based on an auto-encoder architecture with a channel model embedded between the encoder and the decoder. The channel model is constructed using a conditional generative adversarial network (cGAN).

b. Methods:
The neural-modulation training procedure consists of two steps. In the first step, a cWGAN based channel model is trained using a set of vth reads of a single block programmed with given targets. The critic is used solely to train the generator and is then discarded. In the second step, the modulator and demodulator are jointly trained. The modulator translates a block of information symbols into vth targets.

c. Experimental results:
The trained channel model shows good performance in predicting the bit error rate (BER) and generating accurate histograms. The Wasserstein distance between the real and generated histograms is measured to be 6 ± 2 mV, which is comparable to the average distance between two real blocks in the same chip. The predicted BER has an error margin of up to 5%. The modulator extends the number of program/erase (PE) cycles by 25% before the memory device reaches its end-of-life (EOL) point.

**Weaknesses:**

The overhead from programming such models is not adequately discussed

**Questions:**

When quantizing the target, how 60 target is decided (in Appendix)? is it the goal to control BER gap within 0.02? Any rationale?

**Limitations:**

The authors have clearly described the limitations and proposed the next works to tackle.
No negative societal impact.

---

> ### Author Rebuttal · Authors · 2023-08-09
>
> Dear reviewer,
>
> Thank you for carefully reading our manuscript, and pointing out some unclear points.
> Specifically, we plan to further elaborate on the required resources:
>
> First, to ensure no additional latency is introduced due to the modulator, the set of targets for WL_n+1 is calculated while WL_n is programmed. Thus, as long as target calculation is not slower than WL programming, it allows for sufficient time to calculate the targets for WL_n+1 before we begin programming it.
>
> The time it takes to program a single QLC WL is ~2800us. This is because a QLC WL consists of 4 pages, and the time it takes to program a page is ~700us [1]. For every QLC WL, the modulator generates 15X16^2 targets [2]. Consequently, the budget for a single target is ~0.73us, which is equivalent to ~300 clock cycles under an assumption of a HW clock of 400-500Mhz. The cost of a forward pass through the modulator is ~25K multiplication operations (with other operations such as additions and activations adding only a negligible cost). These multiplication operations can be performed in ~200 clock cycles using 128 parallel multipliers – well under our budget.
>
> Finally, implementing the modulator in hardware (the 128 multipliers, as well as other components) requires 40-70K gates. For a typical SSD controller with several million of gates, this added area cost is small.
>
> Below are detailed answers to your question:
>
> Q: When quantizing the target, how 60 target is decided (in Appendix)? is it the goal to control BER gap within 0.02? Any rationale?
>
> A: We have tested the change in the overall BER predicted by the model for different choices of level quantization (number of programming targets). The result, presented in Appendix D.2, reveals that the BER decreases as the number of levels increases, as one might expect. Ideally, we would have a set of levels for each cell in the word line. In practice, each target slightly adds to the latency because it requires an additional verify operation. [As explained in the supplementary materials, the latency overhead of performing successive verify operations with close values is small compared to a single verify operation [3]]. Thus, we based our choice of quantization on the tradeoff between the gain and the latency. As implied from Fig. 1, ~90% of the gain is achieved by increasing the number of targets to about 60, i.e., 4 per level (there is an elbow in the quantization loss curve in the vicinity of 60 levels). The latency introduced by multiplying the number of levels by 4 is reasonable for actual implementation, and hence, it is a natural choice.
>
> [1] C. Kim et al., "A 512-Gb 3-b/Cell 64-Stacked WL 3-D-NAND Flash Memory," in IEEE Journal of Solid-State Circuits, vol. 53, no. 1, pp. 124-133, Jan. 2018, doi: 10.1109/JSSC.2017.2731813.
>
> [2] 16^2 targets are assigned for each level (besides the erase) according to its neighbors.
>
> [3] S. Lee et al., "A 1Tb 4b/cell 64-stacked-WL 3D NAND flash memory with 12MB/s program throughput," 2018 IEEE International Solid - State Circuits Conference - (ISSCC), San Francisco, CA, USA, 2018, pp. 340-342, doi: 10.1109/ISSCC.2018.8310323.

---

> > ### Comment · Reviewer_EdAW · 2023-08-11
> >
> > Thanks for your rebuttal, my questions are well answered.

---

### Official Review · Reviewer_KCDD · 2023-07-07

**Soundness:** 3 good
**Presentation:** 3 good
**Contribution:** 3 good
**Rating:** 6
**Confidence:** 2

**Summary:**

* This paper introduces a unsupervised learning method using GAN for flash memory.

**Strengths:**

* The paper is well written and organized, generally easy to follow.
* To the best of my knowledge, this paper is the first to introduce an unsupervised technique to this kind of problem.
* The core problem is well formatted. The idea of using a cWGAN as channel model looks interesting and intuitive in section 3.
* Experimental results look promising, especially on extending storage life.

**Weaknesses:**

Please see questions

**Questions:**

* On Figure 7, it does give a view of generated WL from the generated model, is there quantized way to measure the quality?
* On 3.5, quantization, to what degree to quantize and is there any effect on the overall performance?
* How much resource does it take to run the modulator? Considering it is to be run on flash controllers.

**Limitations:**

The authors have discussed the limitation of modulator and channel model.

---

> ### Author Rebuttal · Authors · 2023-08-09
>
> Dear reviewer,
>
> We appreciate your thoughtful engagement with our work and the valuable feedback.
> Below are detailed answers to your questions:
>
> Q: On Figure 7, it does give a view of generated WL from the generated model, is there quantized way to measure the quality?
>
> A: In the paper we measured the quality of the generated histograms using two different metrics:
>    1. The BER of the generated WL (compared to the real data histograms we read from the flash). This is the most important metric, as it measures almost exactly the quantity that the modulator aims to optimize.
>    2. The 1-Wasserstein distance between the real and generated distributions (see lines 294-298). We measured a 1-Wasserstein distance of 6 ± 2 mV between the real and generated distributions, which is equivalent to the average 1-Wasserstein distance between two real blocks on the same chip. We note that 6 mV is less than 2% of the width of a level.
>
> Q: On 3.5, quantization, to what degree to quantize and is there any effect on the overall performance?
>
> A: We have tested the change in the overall BER predicted by the model for different choices of level quantization (number of programming targets). The result, presented in Appendix D.2, reveals that the BER decreases as the number of levels increases, as one might expect. Ideally, we would have a set of levels for each cell in the word line. In practice, each target slightly adds to the latency because it requires an additional verify operation. [As explained in the supplementary materials, the latency overhead of performing successive verify operations with close values is small compared to a single verify operation [3]]. Thus, we based our choice of quantization on the tradeoff between the gain and the latency. As implied from Fig. 1, ~90% of the gain is achieved by increasing the number of targets to about 60, i.e., 4 per level (there is an elbow in the quantization loss curve in the vicinity of 60 levels). The latency introduced by multiplying the number of levels by 4 is reasonable for actual implementation, and hence, it is a natural choice.
>
> Q: How much resource does it take to run the modulator? Considering it is to be run on flash controllers.
>
> A: Following is an analysis of the resource consumption of the modulator.
>
> First, to ensure no additional latency is introduced due to the modulator, the set of targets for WL_n+1 is calculated while WL_n is programmed. Thus, as long as target calculation is not slower than WL programming, it allows for sufficient time to calculate the targets for WL_n+1 before we begin programming it.
>
> The time it takes to program a single QLC WL is ~2800us. This is because a QLC WL consists of 4 pages, and the time it takes to program a page is ~700us [1]. For every QLC WL, the modulator generates 15X16^2 targets [2]. Consequently, the budget for a single target is ~0.73us, which is equivalent to ~300 clock cycles under an assumption of a HW clock of 400-500Mhz. The cost of a forward pass through the modulator is ~25K multiplication operations (with other operations such as additions and activations adding only a negligible cost). These multiplication operations can be performed in ~200 clock cycles using 128 parallel multipliers – well under our budget.
>
> Finally, implementing the modulator in hardware (the 128 multipliers, as well as other components) requires 40-70K gates. For a typical SSD controller with several million of gates, this added area cost is small.
>
> [1] C. Kim et al., "A 512-Gb 3-b/Cell 64-Stacked WL 3-D-NAND Flash Memory," in IEEE Journal of Solid-State Circuits, vol. 53, no. 1, pp. 124-133, Jan. 2018, doi: 10.1109/JSSC.2017.2731813.
>
> [2] 16^2 targets are assigned for each level (besides the erase) according to its neighbors.
>
> [3] S. Lee et al., "A 1Tb 4b/cell 64-stacked-WL 3D NAND flash memory with 12MB/s program throughput," 2018 IEEE International Solid - State Circuits Conference - (ISSCC), San Francisco, CA, USA, 2018, pp. 340-342, doi: 10.1109/ISSCC.2018.8310323.

---

### Official Review · Reviewer_yAhR · 2023-07-25

**Soundness:** 3 good
**Presentation:** 3 good
**Contribution:** 4 excellent
**Rating:** 7
**Confidence:** 3

**Summary:**

The authors propose a generative model architecture for compensating for errors in NAND flash memory, with the goal of extending flash memory block lifetime. The architecture consists of a GAN and an autoencoder, where the encoder and decoder represent the memory’s modulator and demodulator respectively. The encoder predicts an optimal $v_{th}$ target for a cell, to which the GAN adds noise simulating errors that occur during writing and storage. The decoder then creates a recovered symbol from the perturbed $v_{th}$ value output by the GAN.

**Strengths:**

-	Memory lifetime and reliability are very important, and an ML approach to model and counteract memory errors sounds promising. In particular, using a GAN to simulate the errors that occur during storage by taking into account the ICI appears novel and is quite clever. A search returned no prior works on unsupervised ML for flash memory modulation.
-	Because the proposed technique modifies only the writing stage, the reading algorithm does not need to change.
-	The work is well-written and easy to understand, especially in explaining the workings of flash memory.
-	The graphics are useful for summarizing and visualizing the proposed method.


**Weaknesses:**

-	Though this appears to be the first unsupervised ML approach to flash memory modulation, no reference is provided to similar prior works on ML and flash memory. A quick search returned several, such as
   - https://ieeexplore.ieee.org/abstract/document/9205258
   - https://iopscience.iop.org/article/10.35848/1347-4065/aba5e0/meta?casa_token=JMOIMn_8RJoAAAAA:EoRkZ4ZeYNlOcGh22YIgLcWTwCxmrbKxvvZF6sbHY3Blc7Nj93sAeaxFCbLdE1aaBL-CgyWbkGY
   - https://ieeexplore.ieee.org/abstract/document/9013031?casa_token=YNITahKT5uEAAAAA:f0HrohD0iLmhUh5PhI_lKSk2GL5rNulHxe3dBhO4nnHo_8iTJdO6mp7uIVi0aFcWrEV9bGBPJA
-	The figure references on lines 315 and 322 (for figures 9a and 9b) appear to be switched.
-	The error bars on figure 9a suggest that the difference in retention time between default and modulated targets is not significant.

**Questions:**

- We customize the $v_th$ target for a given cell, but assume the neighbors use default target values. Would it not be better for the encoder to predict target values for the entire vector of symbols for the WL, then see how the GAN would distort the WL?
- If the GAN is trained based on a channel model, how realistic does this model have to be? For instance, should we make a new GAN for each class of flash device, for each operating condition such as varying temperatures, etc.?
- Section 4.1: Why do you perturb like this? What is the range of the targets?
- In a real-world application, it would be necessary to run the encoder on each write operation. Does this greatly increase writing latency?

---

> ### Author Rebuttal · Authors · 2023-08-09
>
> Dear reviewer,
>
> Thank you for carefully reading our manuscript. We plan to extend the discussion in the revised version to increase the clarity of the paper, incorporate the references and fix the labeling of Fig 9. Specifically, in figure 9a we plot the BER as a function of retention time for several hundred different WL. The curve represents the average BER while the shaded area marks the standard deviation of the test WLs and not an error bar. Indeed, such a plot may not clearly reveal our main achievement. The large overlap between the presented distributions reflects the large variation between the BER of different WLs. As our modulator takes into account the WL index, it assigns optimal targets for each WL. Therefore, it would be better to consider the difference between the BER obtained using the default and modulated targets per WL. We plan to replace the figure with a new one showing the evolution of the BER reduction per WL (the figure is included in the global response PDF).
> We also attach a set of histograms of this quantity measured at different retention times.
> The new figures clearly show BER improvement for all WLs after long retention times. It also demonstrates that the pre-distortion slightly deteriorates the BER at short times. As explained in the paper (lines 316-317), since the BER at these times is overall small, there is no real price for using our modulator.
>
> Below are detailed answers to your questions:
>
> Q: We customize the Vth target for a given cell, but assume the neighbors use default target values. Would it not be better for the encoder to predict target values for the entire vector of symbols for the WL, then see how the GAN would distort the WL?
>
> A: We tested the option of using the predicted target voltages instead of the default values, and found a negligible change to the final targets’ constellation as well as the final BER. For this purpose, we passed the neighbors cell symbols through the modulator before the channel model. We are attaching a figure comparing the target voltages obtained using the two training methods to the global response PDF.
>
> The modulator converges to a marginally different set of targets because there is an order of magnitude difference between the width of each level programmed using the modulator and the level spacing. In other words, the targets assigned to a specific level are typically separated from the default target by no more than 100mV while the difference between neighboring levels are on the order of thousands of mVs. Consequently, the main effect of the neighboring cells arises from the rough position of their level rather than the precise one.  Since training the neural modulation system with the predicted target voltages of the neighbors takes much longer time and requires additional resources, we decided to implement the simplified model.
>
>
> Q: If the GAN is trained based on a channel model, how realistic does this model have to be? For instance, should we make a new GAN for each class of flash device, for each operating condition such as varying temperatures, etc.?
>
> A: New generations of flash devices frequently introduce significant physical and operational differences compared to their ancestors. Therefore, each generation will likely require a new channel model. By contrast, we can incorporate operational conditions such as temperature into the channel model by generalizing it to include additional parameters. For this purpose, both the channel model and modulator should use these additional parameters as input features during the training stage. Consequently, it would be possible to employ a single channel model and modulator for all such conditions.
>
> Q: Section 4.1: Why do you perturb like this? What is the range of the targets?
>
> A: The perturbation experiment was designed to evaluate the channel model, and to reassure us that the GAN reflects the real NAND channel given out-of-sample sets of targets. Specifically, we choose those perturbations from two main reasons:
>
> 1.	They produce a non-trivial level shifts where some of the levels get closer while others depart.
> For example, as shown in Figure 7, levels 1 and 2 depart, while levels 2 and 3 become closer.
>
> 2.	The chosen values of delta test the model at "interesting" parts of the target space, i.e., at targets in which the resulting BER is not too high.
>
> Obviously, there are many other perturbations that would have achieve the same goals.
>
> Q: In a real-world application, it would be necessary to run the encoder on each write operation. Does this greatly increase writing latency?
>
> A: To ensure no additional latency is introduced due to the modulator, the set of targets for WL_n+1 is calculated while WL_n is programmed. Thus, as long as target calculation is not slower than WL programming, it allows for sufficient time to calculate the targets for WL_n+1 before we begin programming it.
>
> The time it takes to program a single QLC WL is ~2800us. This is because a QLC WL consists of 4 pages, and the time it takes to program a page is ~700us [1]. For every QLC WL, the modulator generates 15X16^2 targets [2]. Consequently, the budget for a single target is ~0.73us, which is equivalent to ~300 clock cycles under an assumption of a HW clock of 400-500Mhz. The cost of a forward pass through the modulator is ~25K multiplication operations (with other operations such as additions and activations adding only a negligible cost). These multiplication operations can be performed in ~200 clock cycles using 128 parallel multipliers – well under our budget.
>
> Finally, implementing the modulator in hardware (the 128 multipliers, as well as other components) requires 40-70K gates. For a typical SSD controller with several million of gates, this added area cost is small.
>
> [1] C. Kim et al., "A 512-Gb 3-b/Cell 64-Stacked WL 3-D-NAND Flash Memory"
>
> [2] 16^2 targets are assigned for each level (besides the erase) according to its neighbors.

---

> > ### Comment · Reviewer_yAhR · 2023-08-12
> >
> > Thank you for the further details and graphics, I will increase my rating from 6 to 7 as well.

---

### Author Rebuttal · Authors · 2023-08-09

Dear reviewers,

Following your feedback, we would like to provide a detailed answer for two specific questions, both have relevant figures in the attached PDF:

Q: We customize the Vth target for a given cell, but assume the neighbors use default target values. Would it not be better for the encoder to predict target values for the entire vector of symbols for the WL, then see how the GAN would distort the WL?

A: We tested the option of using the predicted target voltages instead of the default values, and found a negligible change to the final targets’ constellation as well as the final BER. For this purpose, we passed the neighbors cell symbols through the modulator before the channel model. We are attaching a figure comparing the target voltages obtained using the two training methods to the PDF.

The modulator converges to a marginally different set of targets because there is an order of magnitude difference between the width of each level programmed using the modulator and the level spacing. In other words, the targets assigned to a specific level are typically separated from the default target by no more than 100mV while the difference between neighboring levels are on the order of thousands of mVs. Consequently, the main effect of the neighboring cells arises from the rough position of their level rather than the precise one.  Since training the neural modulation system with the predicted target voltages of the neighbors takes much longer time and requires additional resources, we decided to implement the simplified model.

Q: The error bars on figure 9a suggest that the difference in retention time between default and modulated targets is not significant.

A: In figure 9a we plot the BER as a function of retention time for several hundred different WL. The curve represents the average BER while the shaded area marks the standard deviation of the test WLs and not an error bar. Indeed, such a plot may not clearly reveal our main achievement. The large overlap between the presented distributions reflects the large variation between the BER of different WLs. As our modulator takes into account the WL index, it assigns optimal targets for each WL. Therefore, it would be better to consider the difference between the BER obtained using the default and modulated targets per WL. We plan to replace the figure with a new one showing the evolution of the BER reduction per WL (see the attached PDF).
We also attach a set of histograms of this quantity measured at different retention times.

---

### Decision · Program_Chairs · 2023-09-21

**Decision:**

Accept (poster)

**Comment:**

This is a paper on reliable data storage in flash memories. A variety of work in error-correcting codes has sought to handle the various specific errors that arise in these devices, but directly modeling the resulting channel tends to be hard.

This paper proposes a generative model to learn the channel, providing the ability to capture far richer error patterns than manually-built models do.

The basic idea in this paper is simple, but it is well-executed. It joins other papers that show how learned components can provide excellent performance in information theory settings. The reviewers were uniformly positive.